# An Efficient, One-Pot Transamidation of 8-Aminoquinoline Amides Activated by Tertiary-Butyloxycarbonyl

**DOI:** 10.3390/molecules24071234

**Published:** 2019-03-29

**Authors:** Wengang Wu, Jun Yi, Huipeng Xu, Shuangjun Li, Rongxin Yuan

**Affiliations:** 1College of Chemistry, Chemical Engineering and Materials Science of Soochow University, 199 Ren’ai Road, Suzhou, Jiangsu 215123, China; 20155209006@suda.edu.cn; 2Jiangsu Laboratory of Advanced Functional Material, School of Chemistry and Materials Engineering, Changshu Institute of Technology, Changshu 215500, China; xuhp150315217cslg@126.com (H.X.); lisj150315206cslg@126.com (S.L.)

**Keywords:** one-pot reaction, amide C_(acyl)_–N cleavage, palladium, N-heterocyclic carbene, directing group removal

## Abstract

The efficient, one-pot access to the transamidation of 8-aminoquinoline (8-AQ), notorious for its harsh removal conditions, has been widely employed as an auxiliary in C–H functionalization reactions due to its strong directing ability. In this study, the facile and mild Boc protection of the corresponding 8-AQ amide was critical to activate the amide C_(acyl)_–N bond by twisting its geometry to lower the amidic resonance energy. Both aryl and alkyl amines proceeded transamidation in one-pot, user-friendly conditions with excellent yields.

## 1. Introduction

C–H functionalization has emerged as a powerful strategy for building-up new molecules [1,2,3,4]. In the past decades, the growing interest in C–H functionalization has led to increased attention to this paradigm for organic synthesis, which has inspired exceptional advances in this field [5,6,7,8]. The maneuver of directing groups (DGs) in C–H functionalization has been proved to be pivotal to facilitate the desired transformation. Common DGs include ketones [9,10,11], esters [12,13,14], carboxylic acid [15,16,17], amides [18,19,20,21,22], etc. Among them, amides are one of the most frequently used motifs (Scheme 1(A)). Taking advantage of its strong coordination with transition metals, a myriad of contributions have been made, especially with bidentate amide DGs. In 2005, Daugulis [23] introduced 8-aminoquinoline (8-AQ) A2 to C–H functionalization, which has been widely used in Pd- [24,25,26,27], Rh- [28,29,30], Ru- [31], Ni- [32,33,34,35,36,37,38], Cu- [39,40,41], Co- [42,43,44], and Fe-catalyzed [45,46,47,48] C–H activation to enable various reactions.

As a result of electron delocalization (n_N_ to π_C═O_*, ∆H_rot_ of ca. 15–20 kcal/mol in planar amides) [49], the difficulties in uninstalling amide DGs greatly constrain its application (Scheme 1B). Conventional deprotection required stoichiometric amounts of a strong base or acid (Scheme 1(B,B2)), which diminished the functional tolerance [37]. A milder, two-step strategy was therefore widely used, where 8-AQ amides were firstly transformed to tertiary amides by using di-*tert*-butyl decarbonate (Boc_2_O), which was subsequently hydrolyzed by lithium hydroxide and hydrogen peroxide (B2) [50]. Similarly, transesterification strategies by boron trifluoride-diethyl etherate (BF_3_·Et_2_O) were also limited due to the harsh Lewis acid BF_3_ (B3) [37]. Recently, Morimoto and Ohshima reported nickel-catalyzed alcoholysis of unactivated 8-aminoquinoline amides in neutral conditions with excellent functional compatibilities (B3); however, it is mostly limited to alkyl amides [51]. Inspired by the para-methoxyphenyl (PMP) protecting group for amines, Chen installed a methoxy group at the 5 position of 8-AQ for facile removal with ceric ammonium nitrate (CAN) under mild conditions (B4) [52]. Unfortunately, the modified auxiliary is expensive due to its multistep synthesis. Reduction with stoichiometric zirconocene hydrochloride (also known as Schwartz’s reagent) gives the corresponding aldehyde, yet this organometallic reagent is not user friendly [53]. Compared with the booming C–H functionalization reactions, the removal of auxiliary groups is still lagging behind. Meanwhile, Garg [54,55,56,57,58,59,60,61,62,63,64] and Szostak [65,66,67,68,69,70,71,72,73,74,75,76,77,78,79,80,81,82,83] proved that by increasing the sterics of the substitution group, the formation of twisted amides activates the amidyl C_(acyl)_–N bond to favor the insertion of low-valent metals, furnishing the acyl−metal intermediate for further transformations. Inspired by these advances in amidyl C_(acyl)_–N activation, we envisioned that 8-AQ removal could be achieved in this manner after C–H functionalized reactions (Figure 1).

## 2. Results

In order to test our proposal, first, the proper activating group of 8-AQ was chosen. We decided to choose Boc as the activating group for three reasons: (1) The installation of Boc was very moderate, which enhanced the functional group tolerance, with a nearly quantitative yield [84,85,86,87]. (2) 8-aminoquinoline can be easily recycled through deprotection of the *tert*-butyl quinolin-8-ylcarbamate byproduct. (3) Above all, Boc is one of the most thriving activation groups in amidyl C_(acyl)_–N transformation reactions.

Our study was initiated using 3-methyl-*N*-(quinolin-8-yl)benzamide (**1a**) with aniline (**2a**) as model substrates for transamidation in the presence of different bases (Table 1). We started screening with well-developed phosphine ligands. Unfortunately, neither PPh_3_ nor PCy_3_ gave a product. The more σ-donating N-heterocyclic carbenes (NHCs) have revealed high efficiency in inert bond activation, such as C–O [88], C–F [89], as well as amide transformation reactions [63,65,70]. As expected, when using NHC ligands, the desired product was observed. With these promising results, several NHC ligands were examined for transamidation reactions (entries 3–9). Although the combination of Ni(COD)_2_ and SIPr (entry 9) showed a higher yield than (IPr)Pd(cinnamyl)Cl (entry 6), due to the significant difference of stability under air, we decided to use (IPr)Pd(cinnamyl)Cl as the catalyst, which was more user friendly [77]. A higher temperature was critical to improve the yield (entry 10). Neither the strong base sodium *tert*-butoxide (entry 12) nor the organic base 1,8-diazabicyclo[5.4. 0]undec-7-ene (DBU) (entry 13) enhanced the yield. Switching the reaction solvent from THF to MeCN, dioxane or DMF resulted in lower yields (entries 14–16). The control experiment showed that the Pd catalyst was indispensable (entry 17). Finally, we attempted a one-pot transamidation (entry 18) and found that there was no detriment to the yield. The separation of 1a and 4-dimethylaminopyridine (DMAP) was not needed, which enhanced the step efficiency and made the reaction greener.

With the optimized condition in hand, the substrate comparability of the reaction was examined (Figure 2). First, we tested the amine scope. Methyl substitution on both *para* (**3b**) and *ortho* (**3c**) positions gave good yields, although the *ortho* position was more hindered. The electronic effect of the substitution groups did not have a great impact on the yield. Strong electron-donating (**3f**) as well as electron-withdrawing (**3g**) groups were all tolerated. Next, we tested the carboxylic acid part of the amide. The reaction still was not sensitive to the hindrance, as even the bulky *ortho*-phenoxy group gave an excellent yield (**3k**). Heterocycles such as quinoline (**3m**, **3n**) did not corrupt the efficiency of the reaction, while it was noteworthy that even the challenging di-*ortho*-substituted aniline was tolerated (**3m**). More strikingly, aliphatic carboxylic acids, such as ibuprofen-derived species (**3o**), were also workable in this condition.

Next, we tried to expand the scope to aliphatic amines (Figure 3). The reaction took place without Pd catalysts, which was consistent with Szostak’s report [66,76]. For the primary amines, the reaction proceeded well even at room temperature (**3p**, **3q**, **3r**). For the secondary amines, although a higher temperature was required, the yield was still excellent (**3s**, **3t**).

Finally, we scaled up the reaction to the gram scale (Scheme 2A) and found that the yield was consistent with the small-scale reactions. We also isolated the *tert*-butyl quinolin-8-ylcarbamate in 82% yield, from which the Boc deprotection can be removed quantitatively to recover the 8-AQ (Scheme 2B).

For the mechanism, we postulated that the amide bond was activated by Boc via a steric-induced geometry twist, which facilitated the oxidative addition of a Pd–NHC catalyst to generate an acylpalladium intermediate. The more nucleophilic amine then substituted the 8-AQ carbamate via ligand exchanges, which further underwent reductive elimination to form the product while regenerating the Pd(0) species.

## 3. Discussion

To summarize, we reported the Boc-activated transamidation of 8-AQ, which is a widely utilized auxiliary in C–H functionalization. Boc protection of the corresponding 8-AQ amide was critical to activate the amide C_(acyl)_–N bond by twisting its geometry and lowering the amidic resonance energy. Both aryl and alkyl amines proceeded transamidation in excellent yields. The facile and mild Boc activation of the amide C_(acyl)_–N bond as well as the user-friendly one-pot reaction procedure highlight its utilization in the future. This method provides alternative means to remove 8-AQ that is complementary to existing protocols. Future efforts will be directed toward expanding the types of new transformations as well as the different types of directing groups.

## 4. Experimental Section

### 4.1. Representative General Procedure for One-pot Transamidation of 8-Aminoquinoline Amides

8-aminoquinoline amide (0.2 mmol, 60.5 mg), di-*tert*-butyl-dicarbonate (52.4 mg, 1.2 equiv), *N,N-*dimethylpyridin-4-amine (26.9 mg, 1.1 equiv), and acetonitrile (2 mL, 0.1 M) were added to an oven-dried Schlenk pressure vessel equipped with a stir bar. The resulting reaction mixture was stirred at room temperature for 15 h. The solvent was removed under high vacuum after the indicated time. The same reaction vial was charged with aniline (28 mg, 0.3 mmol), potassium carbonate (55 mg, 2.0 equiv), and (IPr)Pd(cinnamyl)Cl (3.9 mg, 3 mol%) (not needed for alkyl amine), placed under a positive pressure of argon, and subjected to three evacuation/backfilling cycles under high vacuum. THF (2 mL, 0.1 M) was added with vigorous stirring at room temperature, and the reaction took place at the indicated temperate (aniline at 110 °C, primary amine at room temperature, and secondary amine at 80 °C) and was stirred for 24 h. The reaction mixture was cooled down to room temperature and diluted with ethyl acetate (10 mL) after the indicated time. The reaction mixture was filtered and concentrated. Purification by chromatography on silica gel (EtOAc/hexanes) afforded the product (yield: 88% (37.2 mg), white solid). Characterization data matched those described above.

### 4.2. Characterization Data for Products ***3a**–**3u*** (Figure 2 and Figure 3)

*3-Methyl-N-phenylbenzamide* (**3a**): White solid. ^1^H-NMR (400 MHz, Chloroform-*d*) δ 7.90 (s, 1H), 7.70–7.59 (m, 4H), 7.40–7.30 (m, 4H), 7.14 (t, *J* = 7.4 Hz, 1H), 2.41 (s, 3H). ^13^C-NMR (101 MHz, CDCl_3_) δ 166.0, 138.7, 138.0, 135.0, 132.5, 129.0, 128.6, 127.8, 124.5, 123.9, 120.2, 21.4.

*3-Methyl-N-(p-tolyl)benzamide* (**3b**): White solid. ^1^H-NMR (400 MHz, Chloroform-*d*) δ 7.77 (s, 1H), 7.68 (s, 1H), 7.63 (dt, *J* = 5.7, 2.2 Hz, 1H), 7.58–7.50 (m, 2H), 7.39–7.31 (dd, *J* = 5.0, 1.9 Hz, 2H), 7.17 (d, *J* = 8.1 Hz, 2H), 2.42 (s, 3H), 2.34 (s, 3H). ^13^C-NMR (101 MHz, CDCl_3_) δ 165.8, 138.6, 135.4, 135.1, 134.1, 132.4, 129.5, 128.6, 127.7, 123.9, 120.2, 21.4, 20.9.

*3-Methyl-N-(o-tolyl)benzamide* (**3c**): White solid. ^1^H-NMR (400 MHz, Chloroform-*d*) δ 7.93 (d, *J* = 8.0 Hz, 1H), 7.76–7.59 (m, 3H), 7.37 (d, *J* = 5.5 Hz, 2H), 7.29–7.19 (m, 2H), 7.11 (t, *J* = 7.5 Hz, 1H), 2.44 (s, 3H), 2.33 (s, 3H). ^13^C-NMR (101 MHz, CDCl_3_) δ 165.8, 138.8, 135.8, 135.0, 132.6, 130.5, 129.2, 128.6, 127.9, 126.9, 125.3, 123.9, 123.1, 21.4, 17.8.

*N-(4-Methoxyphenyl)-3-methylbenzamide* (**3d**): White solid. ^1^H-NMR (400 MHz, Chloroform-*d*) δ 7.69 (d, *J* = 8.8 Hz, 2H), 7.63 (d, *J* = 6.3 Hz, 1H), 7.58–7.50 (m, 2H), 7.40–7.31 (m, 2H), 6.95–6.86 (m, 2H), 3.82 (s, 3H), 2.43 (s, 3H). ^13^C-NMR (101 MHz, CDCl_3_) δ 166.0, 156.5, 138.5, 134.9, 132.3, 131.1, 128.4, 127.8, 124.0, 122.2, 114.1, 55.4, 21.3.

*N-(2-Methoxyphenyl)-3-methylbenzamide* (**3e**): White solid. ^1^H-NMR (400 MHz, Chloroform-*d*) δ 8.53 (d, *J* = 6.6 Hz, 2H), 7.75–7.69 (m, 1H), 7.66 (d, *J* = 6.8 Hz, 1H), 7.37 (d, *J* = 7.3 Hz, 2H), 7.12–6.96 (m, 2H), 6.92 (d, *J* = 7.9 Hz, 1H), 3.93 (s, 3H), 2.44 (s, 3H). ^13^C-NMR (101 MHz, CDCl_3_) δ 165.5, 148.1, 138.6, 135.3, 132.4, 128.6, 127.8, 127.8, 123.9, 123.8, 121.2, 119.8, 109.9, 55.8, 21.4.

*N-(2,5-Dimethoxyphenyl)-3-methylbenzamide* (**3f**): White solid. ^1^H-NMR (400 MHz, Chloroform-*d*) δ 8.56 (s, 1H), 8.29 (d, *J* = 3.0 Hz, 1H), 7.78–7.62 (m, 2H), 7.37 (d, *J* = 7.0 Hz, 2H), 6.83 (d, *J* = 8.9 Hz, 1H), 6.61 (dd, *J* = 8.9, 3.0 Hz, 1H), 3.88 (s, 3H), 3.82 (s, 3H), 2.44 (s, 3H). ^13^C-NMR (101 MHz, CDCl_3_) δ 165.4, 153.9, 142.3, 138.7, 135.2, 132.5, 128.6, 128.5, 127.8, 123.9, 110.7, 108.8, 105.9, 56.3, 55.8, 21.4.

*3-Methyl-N-(3-nitrophenyl)benzamide* (**3g**): Pale yellow solid. ^1^H-NMR (400 MHz, Chloroform-*d*) δ 8.49 (t, *J* = 2.1 Hz, 1H), 8.22 (s, 1H), 8.11 (d, *J* = 8.2 Hz, 1H), 7.98 (d, *J* = 8.2 Hz, 1H), 7.74–7.62 (m, 2H), 7.52 (t, *J* = 8.2 Hz, 1H), 7.37 (d, *J* = 5.3 Hz, 2H), 2.42 (s, 3H). ^13^C-NMR (101 MHz, CDCl_3_) δ 166.2, 148.6, 139.1, 138.9, 133.9, 133.2, 129.9, 128.8, 127.8, 125.9, 124.1, 119.0, 114.9, 21.3.

*3-Methyl-N-(naphthalen-1-yl)benzamide* (**3h**): White solid. ^1^H-NMR (400 MHz, Chloroform-*d*) δ 8.20 (s, 1H), 8.03 (d, *J* = 7.5 Hz, 1H), 7.90 (dt, *J* = 6.5, 2.9 Hz, 2H), 7.84–7.70 (m, 3H), 7.52 (td, *J* = 8.0, 7.3, 4.4 Hz, 3H), 7.41 (d, *J* = 6.0 Hz, 2H), 2.46 (s, 3H). ^13^C-NMR (101 MHz, CDCl_3_) δ 166.5, 138.6, 134.7, 134.1, 132.6, 132.4, 128.7, 128.6, 128.0, 127.6, 126.2, 126.0, 125.9, 125.7, 124.1, 121.4, 120.9, 21.3.

*2-Methyl-N-phenylbenzamide* (**3i**): Pale yellow oil. ^1^H-NMR (400 MHz, Chloroform-*d*) δ 7.65 (d, *J* = 7.9 Hz, 2H), 7.57 (s, 1H), 7.50 (d, *J* = 7.6 Hz, 1H), 7.39 (t, *J* = 8.0 Hz, 3H), 7.29 (t, *J* = 5.3 Hz, 2H), 7.18 (t, *J* = 7.5 Hz, 1H), 2.53 (s, 3H). ^13^C-NMR (101 MHz, CDCl_3_) δ 168.2, 138.0, 136.4, 131.2, 130.2, 129.1, 126.6, 125.9, 124.5, 119.9, 19.8.

*N-(4-Isopropoxyphenyl)-2-methylbenzamide* (**3j**): White solid. ^1^H-NMR (400 MHz, Chloroform-*d*) δ 7.84–7.61 (m, 3H), 7.54 (d, *J* = 8.4 Hz, 2H), 7.44–7.32 (m, 2H), 6.92 (d, *J* = 8.4 Hz, 2H), 4.63–4.47 (m, 1H), 2.45 (s, 3H), 1.36 (d, *J* = 6.0 Hz, 6H). ^13^C-NMR (101 MHz, CDCl_3_) δ 166.2, 154.7, 138.4, 134.9, 132.3, 131.1, 128.4, 127.9, 124.1, 122.3, 116.3, 70.2, 22.0, 21.3.

*N-(3-Nitrophenyl)-2-phenoxybenzamide* (**3k**): Yellow solid. ^1^H-NMR (400 MHz, Chloroform-*d*) δ 9.90 (s, 1H), 8.50 (d, *J* = 2.8 Hz, 1H), 8.35 (d, *J* = 7.9 Hz, 1H), 7.98 (dd, *J* = 14.1, 8.4 Hz, 2H), 7.47 (td, *J* = 10.7, 9.1, 5.7 Hz, 4H), 7.29 (d, *J* = 7.5 Hz, 2H), 7.16 (d, *J* = 8.0 Hz, 2H), 6.90 (d, *J* = 8.3 Hz, 1H). ^13^C-NMR (101 MHz, CDCl_3_) δ 163.1, 155.6, 154.9, 148.6, 139.2, 133.7, 132.6, 130.5, 129.7, 126.0, 125.4, 124.0, 123.0, 119.8, 118.9, 118.2, 115.1.

*N-(2-Fluorophenyl)-4-methylbenzamide* (**3l**): White solid. ^1^H-NMR (400 MHz, Chloroform-*d*) δ 8.40 (d, *J* = 15.6 Hz, 1H), 8.24–8.12 (m, 1H), 7.60–7.46 (m, 3H), 7.31 (t, *J* = 7.6 Hz, 1H), 7.23–7.12 (m, 3H), 2.35 (s, 3H). ^13^C-NMR (101 MHz, Chloroform-*d*) δ 161.2, 160.4 (d, *J* = 246.2 Hz), 135.1, 134.4, 133.6 (d, *J* = 9.4 Hz), 132.3 (d, *J* = 2.0 Hz), 129.5, 125.0 (d, *J* = 3.3 Hz), 121.4 (d, *J* = 11.1 Hz), 120.5, 116.1 (d, *J* = 25.1 Hz).

*N-Mesitylquinoline-2-carboxamide* (**3m**): White solid. ^1^H-NMR (400 MHz, Chloroform-*d*) δ 9.67 (s, 1H), 8.41 (q, *J* = 8.6 Hz, 2H), 8.20 (d, *J* = 8.5 Hz, 1H), 7.95 (d, *J* = 8.2 Hz, 1H), 7.83 (t, *J* = 7.8 Hz, 1H), 7.68 (t, *J* = 7.6 Hz, 1H), 6.99 (s, 2H), 2.34 (s, 3H), 2.32 (s, 6H). ^13^C-NMR (101 MHz, CDCl_3_) δ 162.7, 149.7, 146.5, 137.6, 136.8, 135.2, 131.2, 130.2, 129.8, 129.4, 128.9, 128.0, 127.8, 119.0, 21.0, 18.6.

*N-Phenylquinoline-3-carboxamide* (**3n**): White solid. ^1^H-NMR (400 MHz, Chloroform-*d*) δ 10.27 (s, 1H), 8.41 (q, *J* = 8.5 Hz, 2H), 8.22 (d, *J* = 8.5 Hz, 1H), 8.00–7.78 (m, 4H), 7.68 (t, *J* = 7.6 Hz, 1H), 7.45 (t, *J* = 7.8 Hz, 2H), 7.21 (t, *J* = 7.4 Hz, 1H). ^13^C-NMR (101 MHz, CDCl_3_) δ 162.1, 149.6, 146.3, 137.9, 137.8, 130.3, 129.7, 129.4, 129.1, 128.2, 127.8, 124.3, 119.7, 118.7.

*2-(4-Isobutylphenyl)-N-(4-methoxyphenyl)propenamide* (**3o**): White solid. ^1^H-NMR (400 MHz, Chloroform-*d*) δ 7.31 (d, *J* = 8.6 Hz, 2H), 7.26 (d, *J* = 7.2 Hz, 2H), 7.15 (d, *J* = 7.7 Hz, 2H), 6.97 (s, 1H), 6.80 (d, *J* = 8.6 Hz, 2H), 3.76 (s, 3H), 3.67 (q, *J* = 7.2 Hz, 1H), 2.47 (d, *J* = 7.1 Hz, 2H), 1.95–1.79 (m, 1H), 1.58 (d, *J* = 7.2 Hz, 3H), 0.91 (d, *J* = 6.6 Hz, 6H). ^13^C-NMR (101 MHz, CDCl_3_) δ 172.5, 156.3, 140.9, 138.2, 131.1, 129.7, 127.4, 121.6, 114.0, 55.4, 47.4, 45.0, 30.1, 22.3, 18.5.

*N-(4-Methoxyphenyl)-2-methyl-2-phenylpropanamide* (**3p**): White solid. ^1^H-NMR (400 MHz, DMSO-*d*_6_) δ 9.00 (s, 1H), 7.47 (d, *J* = 8.5 Hz, 2H), 7.42–7.30 (m, 4H), 7.24 (dt, *J* = 6.4, 3.1 Hz, 1H), 6.84 (d, *J* = 8.8 Hz, 2H), 3.70 (s, 3H), 1.55 (s, 6H). ^13^C-NMR (101 MHz, DMSO) δ 174.7, 155.7, 146.5, 132.7, 128.7, 126.7, 126.1, 122.3, 113.9, 55.5, 47.5, 27.3.

*N-Benzyl-3-methylbenzamide* (**3q**): White solid. ^1^H-NMR (400 MHz, Chloroform-*d*) δ 7.61 (s, 1H), 7.57 (s, 1H), 7.39–7.26 (m, 7H), 6.56 (s, 1H), 4.61 (d, *J* = 5.7 Hz, 2H), 2.37 (s, 3H). ^13^C-NMR (101 MHz, CDCl_3_) δ 167.6, 138.4, 138.3, 134.3, 132.2, 128.7, 128.4, 127.9, 127.7, 127.5, 123.9, 44.1, 21.3.

*N-Hexyl-3-methylbenzamide* (**3r**): White solid. ^1^H-NMR (400 MHz, Chloroform-*d*) δ 7.51 (s, 1H), 7.49–7.42 (m, 1H), 7.20 (d, *J* = 4.7 Hz, 2H), 6.30 (s, 1H), 3.44–3.27 (m, 2H), 2.29 (s, 3H), 1.58–1.44 (m, 2H), 1.32–1.16 (m, 6H), 0.87–0.75 (m, 3H). ^13^C-NMR (101 MHz, CDCl_3_) δ 167.8, 138.2, 134.8, 131.9, 128.3, 127.7, 123.8, 40.1, 31.5, 29.6, 26.6, 22.5, 21.3, 14.0.

*N-(Benzo[d][1,3]dioxol-5-yl)-3-methylbenzamide* (**3s**): White solid. ^1^H-NMR (400 MHz, Chloroform-*d*) δ 7.54 (s, 1H), 7.49–7.42 (m, 1H), 7.28–7.24 (m, 2H), 6.81–6.60 (m, 3H), 6.26 (s, 1H), 3.64 (q, *J* = 6.6 Hz, 2H), 2.83 (t, *J* = 6.9 Hz, 2H), 2.37 (s, 3H). ^13^C-NMR (101 MHz, CDCl_3_) δ 167.7, 147.9, 146.2, 138.4, 134.6, 132.6, 132.1, 128.4, 127.6, 123.7, 121.7, 109.1, 108.3, 100.9, 41.3, 35.4, 21.3.

*N-Cyclohexyl-3-methylbenzamide* (**3t**): White solid. ^1^H-NMR (400 MHz, Chloroform-*d*) δ 7.57 (s, 1H), 7.55–7.48 (m, 1H), 7.35–7.27 (m, 2H), 5.93 (s, 1H), 4.05–3.90 (m, 1H), 2.40 (s, 3H), 2.10–1.96 (m, 2H), 1.84–1.56 (m, 4H), 1.52–1.35 (m, 2H), 1.29–1.18 (m, 2H). ^13^C-NMR (101 MHz, CDCl_3_) δ 166.8, 138.2, 135.1, 131.8, 128.3, 127.6, 123.8, 48.6, 33.1, 25.5, 24.9, 21.3.

*3-Methyl-N-(octan-2-yl)benzamide* (**3u**): Colorless oil. ^1^H-NMR (400 MHz, Chloroform-*d*) δ 7.59 (s, 1H), 7.54 (t, *J* = 4.6 Hz, 1H), 7.26 (d, *J* = 4.6 Hz, 2H), 6.53 (s, 1H), 3.36 (td, *J* = 6.2, 2.0 Hz, 2H), 2.35 (s, 3H), 1.65–1.48 (q, *J* = 6.1 Hz, 1H), 1.42–1.23 (m, 8H), 0.98–0.80 (m, 6H). ^13^C-NMR (101 MHz, CDCl_3_) δ 167.9, 138.2, 134.9, 131.9, 128.3, 127.7, 123.8, 43.0, 39.4, 31.0, 28.9, 24.3, 23.0, 21.3, 14.0, 10.8.

*Tert-Butyl quinolin-8-ylcarbamate* (**3v**): Brown solid. ^1^H-NMR (400 MHz, Chloroform-*d*) δ 9.05 (s, 1H), 8.82 (dd, *J* = 4.1, 2.1 Hz, 1H), 8.44 (d, *J* = 7.6 Hz, 1H), 8.16 (d, *J* = 8.3 Hz, 1H), 7.54 (t, *J* = 8.0 Hz, 1H), 7.45 (d, *J* = 7.8 Hz, 2H), 1.60 (s, 9H).^13^C-NMR (101 MHz, CDCl_3_) δ 152.94, 148.00, 138.24, 136.30, 135.18, 128.07, 127.38, 121.56, 120.18, 114.42, 80.45, 28.43.

*8-Aminequinoline* (**3x**): Pale yellow solid. ^1^H-NMR (400 MHz, DMSO-*d*_6_) δ 8.73 (dt, *J* = 3.7, 1.8 Hz, 1H), 8.18 (dt, *J* = 8.3, 1.8 Hz, 1H), 7.46 (ddd, *J* = 8.3, 4.2, 1.9 Hz, 1H), 7.30 (td, *J* = 7.9, 1.8 Hz, 1H), 7.06 (dd, *J* = 8.1, 2.2 Hz, 1H), 6.87 (d, *J* = 7.6 Hz, 1H), 5.94 (s, 2H). ^13^C-NMR (101 MHz, DMSO) δ 147.43, 145.68, 137.84, 136.31, 129.02, 128.07, 121.91, 114.12, 109.09.

## 5. Conclusions

In summary we reported a mild, one-pot Pd catalyzed transamidation of 8-aminoquinoline. The mind and facile Boc protection activate the amide C_(acyl)_–N bond by twisting its geometry to lower the amidic resonance energy for facile Pd oxidative addition. This strategy gives a new choice in removal of directing groups after the C-H functionalization.

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
