# Peer review of "An Efficient, One-Pot Transamidation of 8-Aminoquinoline Amides Activated by Tertiary-Butyloxycarbonyl"

_molecules, 2019, doi:10.3390/molecules24071234_

Round 1

Reviewer 1 Report

The manuscript reports an efficient, one-pot transamidation of 8-2 aminoquinoline amides.

Boc protection of the corresponding 8-AQ amide was critical to activate the amide C(acyl)-N bond by twisting its geometry and lower the amidic resonance  energy. Both aryl and alkyl amines proceeded transamidation in excellent yields. Overall the research is well described and the supporting information is sufficient to demonstrate that the target products was indeed prepared, however, there are some concerns regarding the synthetic procedure.

I think that the manuscript can be accepted after major revisions:

·         Authors claim that in this protocol they used a” facile and mild Boc activation of the amide C(acyl)-N bond”.However, in literature are reported many methods that are milder and simpler respect to that reported in the manuscript. Why authors did not try a green procedure for this boc protection? In any case they should at least cite in the introduction section some relevant procedure (RSC Advances, 2015, 5(78), pp. 63407-63420; RSC Advances, 2015, 5(24), pp. 18751-18760; Asian Journal of Chemistry, 2017, 29(6):1313-1316; Tetrahedron Letters , Volume 49, Issue 16, 14 April 2008, Pages 2527-2532).

·         In addition they describe their procedure in the experimental section (lines 134-148) and after the boc protection procedure they only remove the solvent under reduced pressure and proceed to the second step in the same reaction vessel …but the excess DMAP is still present? How do they eliminate this base?

·         In the manuscript figures have to be mentioned in the main text.

·         The manuscript requires significant proof reading and revision to improve the quality of English.

Author Response

The manuscript reports an efficient, one-pot transamidation of 8-2 aminoquinoline amides.

Boc protection of the corresponding 8-AQ amide was critical to activate the amide C(acyl)-N bond by twisting its geometry and lower the amidic resonance  energy. Both aryl and alkyl amines proceeded transamidation in excellent yields. Overall the research is well described and the supporting information is sufficient to demonstrate that the target products was indeed prepared, however, there are some concerns regarding the synthetic procedure.

I think that the manuscript can be accepted after major revisions:

Point 1: Authors claim that in this protocol they used a” facile and mild Boc activation of the amide C(acyl)-N bond”.However, in literature are reported many methods that are milder and simpler respect to that reported in the manuscript. Why authors did not try a green procedure for this boc protection? In any case they should at least cite in the introduction section some relevant procedure (RSC Advances, 2015, 5(78), pp. 63407-63420; RSC Advances, 2015, 5(24), pp. 18751-18760; Asian Journal of Chemistry, 2017, 29(6):1313-1316; Tetrahedron Letters , Volume 49, Issue 16, 14 April 2008, Pages 2527-2532).

Response 1: The reason we chose this Boc protection protocol was it was a mature method in previous 8-AQ transformation (J. Am. Chem. Soc. 2013, 135, 12135−12141; J. Am. Chem. Soc., 2018, 140, 3542–3546.) But we are grateful for this kind suggestion by the reviewer and we cite these literature which also support that Boc installation was “facile and mild”.

Point 2: In addition they describe their procedure in the experimental section (lines 134-148) and after the boc protection procedure they only remove the solvent under reduced pressure and proceed to the second step in the same reaction vessel …but the excess DMAP is still present? How do they eliminate this base?

Response 2: The excess DMAP was still in the reaction. DMAP did not diminished the yield as we showed in Table 1 entry 10 and 18.

Point 3:  In the manuscript figures have to be mentioned in the main text.

Response 3: We mentioned figures in the main text.

Point 4: The manuscript requires significant proof reading and revision to improve the quality of English.

Response 4: We have polished the paper again with native speaker.

We thank the reviewer for the approval of publication.

Reviewer 2 Report

The authors report an efficient Pd-catalyzed transamidation of 8-aminoquinoline amide. The results are very fine ones and this reviewer believes that the work is worth for publication on this journal. However, several errors should be corrected prior to the publication.

(1)  No compound number is illustrated in Scheme 1 and Figure 1.  

(2)  Quality of Figure in Table 1 should be improved, since space is too narrow between equation and chemical formula of ligands.

(3)  Scheme 2 should be revised. Chemical structure of 3v, 3x are wrong.

Author Response

Response to Reviewer 2 Comments

The authors report an efficient Pd-catalyzed transamidation of 8-aminoquinoline amide. The results are very fine ones and this reviewer believes that the work is worth for publication on this journal. However, several errors should be corrected prior to the publication.

Point 1: No compound number is illustrated in Scheme 1 and Figure 1. 

Response 1: We added compound number in Scheme 1 and Figure 1.

Point 2: Quality of Figure in Table 1 should be improved, since space is too narrow between equation and chemical formula of ligands.

Response 2: We added a frame to make it more clarify.

Point 3: Scheme 2 should be revised. Chemical structure of 3v, 3x are wrong.

Response 3: We made these changes.

We thank the reviewer for the approval of publication.

Round 2

Reviewer 1 Report

The manuscript reports an efficient, one-pot transamidation of 8-2 aminoquinoline amides.

 I think that the manuscript can be accepted in present form.